# The Role of Glycosylation in Melanoma Progression

**DOI:** 10.3390/cells10082136

**Published:** 2021-08-19

**Authors:** Chiara De Vellis, Silvia Pietrobono, Barbara Stecca

**Affiliations:** Tumor Cell Biology Unit, Core Research Laboratory, Institute for Cancer Research and Prevention (ISPRO), Viale Pieraccini 6, 50139 Florence, Italy; chiaradevellis@gmail.com (C.D.V.); silvia.pietrobono@ittumori.it (S.P.)

**Keywords:** melanoma, metastasis, immune evasion, glycosylation, sialylation, fucosylation, glycan branching

## Abstract

Malignant melanoma is the most aggressive form of skin cancer, which originates from the malignant transformation of melanocytes, the melanin-producing cells of the skin. Melanoma progression is typically described as a stepwise process in which metastasis formation ensues late during disease. A large body of evidence has shown that the accumulation of genetic and epigenetic alterations drives melanoma progression through the different steps. Mortality in melanoma is associated with metastatic disease. Accordingly, early-stage melanoma can be cured in the majority of cases by surgical excision, while late-stage melanoma is a highly lethal disease. Glycosylation is a post-translational modification that involves the transfer of glycosyl moieties to specific amino acid residues of proteins to form glycosidic bonds through the activity of glycosyltransferases. Aberrant glycosylation is considered a hallmark of cancer as it occurs in the majority of tumor types, including melanoma. The most widely occurring glycosylation changes in melanoma are represented by sialylation, fucosylation, and N- and I-glycan branching. In this review, we discuss the role of glycosylation in melanoma and provide insights on the mechanisms by which aberrant glycosylation promotes melanoma progression through activation of invasion and metastasis, immune evasion and cell proliferation.

## 1. Introduction

Malignant melanoma is a highly aggressive tumor that accounts for about 5% of all skin cancers, but is responsible for more than 80% of skin cancer-related deaths [1]. Melanoma is the fifth most common form of cancer in adults, and its incidence has been rising worldwide [2]. Melanoma originates from the malignant transformation of melanocytes, the melanin-producing cells of the skin, eye, mucosal epithelia and meninges. Pigments of melanin play a protective role against ultraviolet (UV) radiations, preserving cells against DNA damage [3].

It is a common belief that melanoma progression is a stepwise process in which normal melanocytes evolve to metastatic disease through a series of linear steps. The first phenotypic change in melanocytes is the development of benign nevi, which consists of a population of melanocytes with aberrant proliferation that do not progress due to cellular senescence. Overcoming senescence is one of the critical switches that can lead to dysplastic nevi, which can subsequently progress to a superficial spreading stage (radial growth phase) that is confined in the epidermis. When melanoma cells acquire the ability to invade the dermis (vertical growth phase), they can metastasize and spread throughout the body. The accumulation of genetic and epigenetic changes is thought to drive progression through these steps [4]. Although this model describes the natural history of a portion of melanoma, experimental, clinical and epidemiological data indicate that not all melanomas arise in such a stepwise manner, and metastatic disease can arise in patients without overt primary melanoma, suggesting that melanoma progression might be more complex and less linear [5]. Mortality in melanoma is associated with metastatic disease. In general, the majority of patients with early-stage (I and II) melanoma have an overall favorable prognosis. Patients with stage III melanoma have a heterogeneous prognosis, and those with stage IV melanoma have a poor prognosis. The number of metastasis is also a negative prognostic factor. For instance, melanoma patients with three or more sites of metastasis have a high probability to die within one year [1].

Melanoma is the tumor with the highest mutational burden [6]. Common genetic alterations associated with melanoma include mutually exclusive mutations in BRAF (50–60%), NRAS (20–25%) and NF1 (14%) [7,8]. These mutations drive hyperactivation of the mitogen-activated protein kinases (MAPK) extracellular signal-regulated kinase 1 and 2 (ERK1/2) [9], which promotes tumor cell growth.

Until recently, there were no effective therapies for patients with advanced melanoma. However, significant advances in the understanding of the biological and molecular basis of melanoma in the last decade have led to the development of novel treatments, including targeted therapy and immunotherapy. Activation of oncogenic BRAF has provided the basis for targeted therapy with specific inhibitors of mutant BRAF and the downstream kinase MEK1/2 [10,11]. Despite the remarkable responses of BRAF mutant melanoma patients to BRAF and MEK inhibitors, primary resistance or development of acquired resistance to these targeted therapies remain a significant issue. The combination of BRAF inhibitors and MEK inhibitors such as trametinib delays the development of resistance. Immune checkpoint inhibitors against cytotoxic T-lymphocyte antigen 4 (CTLA-4) or programmed death receptor-1 (PD-1)/programmed death-ligand 1 (PD-L1) have shown more durable responses. One in two patients with metastatic melanoma are alive five years after diagnosis when treated with combination immunotherapy [12]. The major weaknesses of immunotherapy are the lack of predicted biomarkers of response and toxicity, which can be challenging in some patients and may limit treatment [13]. Despite these advances, additional therapeutic approaches are needed for patients, resistant or not, responding to the currently available targeted and immune treatments.

It is well known that when normal cells evolve to a neoplastic state, they acquire a number of capabilities that make them tumorigenic and ultimately malignant. Aberrant glycosylation is considered a hallmark of cancer that plays a critical role in several cellular functions, including proliferation and cell death, cell adhesion, migration, invasion and angiogenesis [14]. Glycosylation is an enzymatic process that links glycans to proteins or lipids. It is the most common post-translational modification (PTM), and one of the most complex. The most widely occurring glycosylation events in melanoma are sialylation, fucosylation, and N- and I-glycan branching. In this review, we discuss how these events of aberrant glycosylation play a role in melanoma progression.

## 2. Aberrant Glycosylation in Melanoma

Glycosylation is a PTM that leads to the covalent attachment of sugar moieties on nascent proteins [15]. The formation of glycosidic bonds is dynamically regulated by various enzymes, namely glycosyltransferases and glycosidases. In glycoproteins, the linkage of the glycan chains to their polypeptide backbone typically interests the nitrogen of asparagine (N-glycans) or the oxygen of serine or threonine (O-glycans) [16]. In physiological conditions, glycosylation is crucial for a wide range of biological processes such as protein folding, activation and degradation, and for mediating cellular interactions with the surrounding extracellular matrix (ECM) [17].

Aberrant expression of glycans is related to cancer development. Altered protein glycosylation is involved in the activation of oncogenic signaling pathways, loss of cell−cell adhesion, cell migration and invasion, dissemination of primary tumor cells throughout the body and acquisition of pro-metastatic behavior [16,17].

In this review, we elucidate the role of key glycosyltransferases in melanoma progression. Cumulative findings show that glycosylation is mostly responsible for a pro-invasive and/or pro-metastatic phenotype in melanoma [18]. This occurs through the abnormal expression of selected glycosyltransferases, such as *N*-glycan branching enzymes, *N*-acetylglucosaminyltransferases V (GnT-V), *N*-acetylglucosaminyltransferases III (GnT-III), sialyltransferases (ST3GAL1 and ST6GAL1), α-1,2 fucosyltransferases (FUT1 and FUT2) and α-1,6 fucosyltransferase (FUT8) [17,19,20] (Figure 1).

Glycosylation is one of the most recurrent PTMs in melanoma, with GnT-III, GnT-V, ST3GAL1, ST6GAL1 and FUT8 as the most aberrantly expressed glycosyltransferases, and integrins, epidermal growth factor receptor (EGFR), cell adhesion molecule L1 (L1CAM) and the receptor tyrosine kinase AXL among the main glycosylated substrates [15]. Accumulating evidence indicates that cell membrane immune checkpoint proteins, such as PD-L1, are glycosylated with heavy N-linked glycan moieties in human cancers, including melanoma. N-linked glycosylation of PD-L1 maintains its protein stability and interaction with its cognate receptor, PD-1, promoting evasion of T-cell immunity [29,30].

Alterations in the carbohydrate structures can affect proliferation, migration, immune escape and apoptosis [15]. A study by Laidler and colleagues revealed that N-glycan changes of cell surface proteins could contribute to the migration process in melanoma [31]. Authors showed that metastatic melanoma cells possess different adhesion properties to ECM components in comparison to the primary melanoma. Moreover, they found a significant correlation between increased α2,6-sialylation and adhesion abilities of metastatic melanoma cells [31]. In line with these observations, Kremser and colleagues identified N-glycosylation alterations on α3β1 and α5β1 integrins of melanoma metastatic cells (WM9 and WM239), including sialylated and fucosylated N-glycans with branches and polylactosamine (plylacNAc) chains. These modifications have been related with increased melanoma cell motility, impacting cell-ECM interactions [32].

Another work from Thies and colleagues showed that N-linked glycosylation could represent a valid prognostic marker for the metastatic phenotype of melanoma. Through a 10-year retrospective study, authors demonstrated that N-acetyl-galactosamine and/or -glucosamine residues are of functional importance for metastasis formation in melanoma, while neither β1-6 branched oligosaccharides nor sialic acid residues appeared to play a pivotal role in this context, although they correlated with invasive behavior in other cancer types [33]. In keeping with that, More and colleagues demonstrated that poly-N-acetyl lactosamine (polyLacNAc) structures on N-glycans promote lung metastasis formation in a B16F10 model of melanoma [34]. Authors showed that polyLacNAc interacts with galectin-3, which is highly expressed on the surface of lung and vascular endothelial cells, and that inhibition of polyLacNAc synthesis or competitive inhibition of its interaction with galectin-3 inhibited lung-specific metastasis in this experimental setting [34].

Recent findings provided insights into the role of β-1,3-galactosyl-*O*-glycosyl-glycoprotein β-1,6-N acetylglucosaminyltransferases 3 (GCNT3) in mediating melanoma progression [28]. The authors showed that silencing of GCNT3 suppresses migration and invasion of melanoma cells. Mechanistically, GCNT3 led to increased S100A8/A9-mediated cell migration and invasion through the stabilization and activation of melanoma cell adhesion molecule (MCAM) (Figure 1). Consistent with these observations, GCNT3 was found overexpressed in highly metastatic melanoma, where it positively correlates with MCAM expression [28]. In another report, aberrant N-glycosylation of the adhesion receptor SHPS-1 (also known as SIRPα1) has been related to the resistance of B16F10 melanoma cells to CD47-mediated negative regulation of motility [35]. Ogura and colleagues characterized and compared SHPS-1 expressed in the mouse nontumorigenic melanocytes Melan-a and the melanoma cell line B16F10. Results suggested that the soluble SHPS-1 ligand (CD47-Fc fusion protein) binds to Melan-a nontumorigenic melanocytes but not to B16F10 melanoma cells. Treatment of these cells with 1-deoxymannojirimycin, which impedes N-glycan processing, reverts the ability of these cells to bind CD47-Fc in vitro, confirming that aberrant N-glycosylation compromises the ability of SHPS-1 to bind CD47-Fc in B16F10 cells. Importantly, CD47-Fc inhibited the migration of Melan-a cells but not that of B16F10 cells. Hence, these results provide evidence that aberrant N-glycosylation of SHPS-1 may contribute to the highly motile phenotype of malignant melanoma [35]. Finally, Kinslechner and colleagues showed that the scavenger receptor class B type I (SR-B1), a high-density lipoprotein (HDL) receptor, is linked with increased cellular glycosylation in metastatic melanoma. Indeed, knockdown of SR-B1 decreased migration and invasion of melanoma cells and reduced xenograft tumor growth. Furthermore, lectin immunoprecipitation revealed a marked reduction of glycosylated STAT5 in the SR-B1 knockdown groups. As STAT5 mediates epithelial-to-mesenchymal transition (EMT), SR-B1-related glycosylation is critical for maintaining metastatic phenotypes of melanoma [36].

Unlike N-glycosylation, little is known about the role of O-glycosylation in promoting melanoma progression. A study by Wagner and colleagues reported a link between O-glycosylation and apoptotic signaling in several tumors, including melanoma [37]. The authors found that the peptidyl O-glycosyltransferase GALNT14 catalyzes the glycosylation of the death-receptors TRAIL-R1/DR4 and TRAIL-R2/DR5, increasing their affinity for the proapoptotic ligand Apo2L/TRAIL. Consistently, they demonstrated that GALNT14 knockdown decreased ligand-induced receptor clustering, with subsequent inhibition of the extrinsic apoptotic signaling, highlighting the potential use of GALNT14 as a predictive biomarker for Apo2L/TRAIL-based cancer therapy [37]. However, despite the importance of O-glycosylation in several tumors, the role of O-glycans in melanoma progression has not been elucidated yet and requires more in-depth studies.

## 3. Sialylation

Sialylation is an enzymatic process that consists in the covalent addition of sialic acid to the terminal position of glycan chains on glycoproteins and glycolipids. The synthesis of sialylated glycans is catalyzed by the sialyltransferase family, which consists of 20 subtypes in humans. All of them can produce both N-linked and O-linked glycans, which are involved in several cellular processes. Golgi-located or membrane-bound sialyltransferases use cytidine monophosphate N-acetylneuraminic acid (CMP-Neu5Ac) as a donor to catalyze the formation of a glycosidic linkage between C2 of the acid sialic from the donor and C3, C6 or C8 hydroxyl of a glycan acceptor. Accordingly, sialyltransferase subtypes are named ST3, ST6 or ST8. These, in turn, can be further classified based on the acceptor sugar, which can be galactose (Gal), *N*-acetylgalactosamine (GalNAc), or a further sialic acid (Sia) moiety [38]. Expression of the 20 sialyltransferases is regulated in a context-dependent manner. Each of them is characterized by substrate specificity, although with some degree of redundancies [39]. Sialylation is dynamically regulated by another set of enzymes, neuraminidases or sialidases, which consist of four human enzymes (NEU1-4) that cleave sialic acid from the terminal ends of glycan chains, thereby regulating cell-surface sialylation. Hypersialylation can be the effect of upregulation of sialyltransferases, downregulation of neuraminidases or a combination of both [40]. Accumulation of sialic acid on the cell surface promotes tumor metastasis by enhancing immune evasion and stimulating migration, invasion and angiogenesis [41,42,43].

The link between melanoma progression and aberrant sialylation is well established. Increased α2,3-sialylation is associated with a more aggressive phenotype in melanoma cells. In particular, the presence of aberrant sialoglycoconjugates on α5β1 integrin has been shown to correlate with a more aggressive phenotype in melanoma cell lines [44]. A recent study from our group reported that the expression of the α 2,3-sialyltransferase ST3GAL1 correlates with melanoma progression and highlighted the critical role of ST3GAL1 in driving melanoma metastasis [20]. Silencing of this sialyltransferase suppressed melanoma migration and invasion and reduced the ability of aggressive melanoma cells to enter the bloodstream, colonize distal organs and survive in the metastatic environment. Further, this study shed light on the regulation of ST3GAL1 in melanoma cells. It was found that the oncogenic transcription factors GLI1 (one of the final mediators of the Hedgehog pathway) and SOX2 (a known pluripotency transcription factor) co-regulate ST3GAL1 transcription. Moreover, the tyrosine kinase receptor AXL was identified as a ST3GAL1 substrate, which is fundamental for the pro-invasive effects of ST3GAL1 in melanoma cells. Collectively, these results unveil the role of the SOX2/GLI1-ST3GAL1-AXL axis in melanoma progression [20] (Figure 1). Another report provided insight into the role of ST6GAL1 in melanoma. Indeed, the depletion of sialylated glycoconjugates by either enzymatic desialylation or silencing of ST6GAL1 decreased the adhesion on both extracellular matrix (ECM) and basement membrane (BM) components and reduced invasiveness of murine B16 melanoma cells [45]. However, the mechanism by which these α2,6-linked oligosaccharides modulate adhesion on ECM and BM components has not been investigated.

Recent evidence highlighted the role of sialic acids on the surface of cancer cells to protect them from destruction by the immune system [46] in several types of cancer, including melanoma. Perdicchio and colleagues have first shown that hypersialylation of B16 melanoma cells positively affects tumor growth by facilitating the escape from the immune system at multiple levels [47]. Reduction of sialic acids on B16 cells by silencing the CMP-sialic acid transporter Slc35a1 slowed down tumor growth in vivo and promoted an anti-tumor immune response, boosting the tumor infiltration of effector T cells and decreasing the frequency of T regulatory cells. The authors also provided evidence that the altered Treg/Teff balance in this model depends on the increased influx and activity of natural killer (NK) cells, as their depletion abolished the induction of anti-tumor immunity. Thus, reducing sialylation could provide a therapeutic strategy to make melanoma cells more susceptible to immune attack [47].

Dendritic cells (DCs) represent a class of immune cells which offer good prospects for anti-cancer immunotherapy [48]. However, human DCs display a high content of sialic acids, which inhibits their maturation and co-stimulation ability. Silva and colleagues have shown that desialylation of DCs improves their ability to elicit T cell-mediated anti-tumor activity [49]. Indeed, desialylation of human DCs induced their maturation, increased expression of major histocompatibility complex I and II (MHC-I and -II) and co-stimulatory molecules, and stimulated IL-12 secretion. Furthermore, desialylated DCs displayed increased peptide presentation through MHC-I, leading to activation of CD8+ cytotoxic T cells. Accordingly, desialylated DCs in co-culture with OVA-expressing B16 melanoma cells improved T cell-mediated cytotoxicity against tumor cells. Therefore, sialidase treatment of DCs might represent a novel tool to improve the efficacy of DC-based anti-cancer vaccines [49].

Although elevated immunoglobulin (IgG) levels in patients with cancer are believed to be the result of increased expression of B cell-derived anti-tumor antibodies, a lot of evidence indicates that IgG in the tumor microenvironment (TME) has pro-tumorigenic activity by blocking T cell-mediated tumor cytotoxicity or pro-inflammatory activity [50]. A recent study by Wang and colleagues suggested that cancer cells can secrete IgG into the TME [51]. Interestingly, authors found that Cancer-IgG could be involved in tumor immune evasion by inhibiting proliferation and reducing the number of effector CD4+ and CD8+ T cells. Mechanistically, Cancer-IgG binds to the sialic acid receptors called Siglecs expressed on effector T cells through a unique sialylation of the CH1 domain, promoting the maintenance of an immunosuppressive TME. Altogether, these findings indicate that sialylated Cancer-IgG may function as potential immune checkpoint proteins and mediate immune evasion in melanoma and other types of cancer [51].

A recent study demonstrated the implication of Siglec-1 in melanoma lymph node (LN) metastasis [21]. Singh and Choi reconstructed early LN colonization events by implanting mice with GFP-expressing B16 melanoma cells. They showed that Siglec1-expressing subcapsular sinus (SCS) macrophages provide anchorage to pioneer metastatic cells. In vitro cocultures confirmed that interactions between hypersialylated B16-GFP and Siglec1-expressing HEK-293T cells drive the proliferation of metastatic cells. Transcriptomic profiles of Siglec1-interacting cancer cells versus non-Siglec1-interacting cancer cells revealed enrichment in positive regulators of cell cycle progression. CRISPR-mediated knockout of the ST3GAL3 sialyltransferase in B16-GFP melanoma cells reduced α2,3-linked sialylation and compromised the metastatic ability of tumor cells. This study suggests that the interaction between Siglec1-expressing SCS macrophages and metastatic cells regulates cell cycle progression and facilitates an efficient metastatic colonization of melanoma cells [21] (Figure 1).

Haas and colleagues provided evidence that melanoma-intrinsic hypersialylation suppresses effector functions of Siglec-9+ CD8+ T cells in the TME [23]. RNA-sequencing data from the TCGA melanoma database revealed that the sialyltransferases ST3GAL5, ST6GALNAC2 and ST3GAL6 are consistently expressed in the investigated melanoma samples. As a consequence, enzymatic digestion of Siglec-9 ligands on target cells by neuraminidase only enhanced the cytotoxicity of the Siglec-9-expressing cells. Thus, key effector functions of CD8+ T cells are abrogated by Siglec-9 engagement, which is related to the phosphorylation of the inhibitory protein phosphates SH-P1 in vitro. The reported findings show that selected sialyltransferases are implicated in the biosynthesis of Siglec-9 ligands, which exert an immunosuppressive role in melanoma progression [23] (Figure 1).

Another report identified CD43, a mucin-like transmembrane protein, as a further promising immunotherapeutic target for melanoma [52]. Sialylated CD43 (CD43s) is expressed by hematologic malignancies, and it is recognized by the monoclonal antibody AT1413 [52]. De Jong and colleagues immunoprecipitated CD43s from melanoma cells, confirming that AT1413 could bind to CD43s in melanoma [53]. AT1413 was unable to affect growth of melanoma cells in vivo, but induced antibody-dependent cellular cytotoxicity against short-term patient-derived melanoma cells. In order to increase the efficacy of AT1413, authors generated two different formats of bi-specific T-cell engagers (TCEs). These antibodies increased T-cell cytotoxicity against melanoma cells in vitro with a certain potency. In line with this, TCEs treatment in mice harboring a human immune system grafted with A375 melanoma cells induced a strong tumor growth inhibition and T-cell engagement, indicating that CD43s-binding receptors on T cells may orchestrate immune evasion during melanoma progression [53]. In brief, these findings provide novel T-cell engaging antibodies that might be combined with current immunotherapies in melanoma.

## 4. Fucosylation

Fucosylation is a type of glycosylation that catalyzes the attachment of fucose sugar units to a molecule. Fucosylation is performed by fucosyltransferase (FUT) enzymes. Fucosylation of glycoproteins is one of the most important features that mediate several specific biological functions in normal and cancer cells. FUT is a group of enzymes that catalyze the incorporation of fucose from activated nucleotide donor GDP-fucose to the reducing end of complex glycans in a linkage-specific manner. These enzymes are widely expressed in many different tissues [54]. Thirteen fucosyltransferase genes have been identified in the human genome. These can be further classified into three subfamilies: α-1,2 FUT, α-1,3/4 FUT and α-1,6 FUT [55,56]. Fucosylation can be divided into terminal and core fucosylation. Among FUT enzymes, FUT8 is an α-1,6 fucosyltransferase and the only FUT responsible for core fucosylation on N-glycoproteins, as most of the other fucosyltransferases are functionally redundant [55,57,58,59]. In cancer, fucosyltransferases play an important role in the biosynthesis of tumor-associated antigens, including Lewis (Le)^a^ and (Le)^b^, sialyl Lewis A (sLe)^a^ and sialyl Lewis X (sLe)^x^, as well as the H blood group antigen [54].

One of the first pieces of evidence that fucosyltransferases might be implicated in melanoma progression came from the finding that the expression levels of FUT1 and FUT4 mRNA are significantly higher in metastatic melanoma cell lines (A375, WM9, WM239) compared to primary melanoma cells (WM35) [60]. LTA (*Lotus tetragonolobus agglutinin*), the lectin that specifically recognizes fucose residue of glycans and L-fucose, reduced adhesion to fibronectin and collagen IV of all primary and metastatic cell lines and slightly decreased proliferation of metastatic but not primary melanoma cell lines [60]. A more recent report confirmed the importance of FUT4 in migration and invasion of human melanoma cells that occur through the activation of the PI3K/AKT signaling pathway [27] (Figure 1).

Other studies pointed toward a role of FUT4 in regulating melanoma cell growth. It has been shown that NF-κB/p65-dependent transcriptional regulation of FUT4 modulates human melanoma cell proliferation [61]. Interestingly, administration of the ginsenoside Rg3, an herbal medicine with anti-tumor activity, suppresses the growth of human melanoma xenografts by decreasing the expression levels of FUT4 and p65 in vivo [61]. Another report showed that Rg3 inhibits melanoma cell growth through inhibition of EGFR phosphorylation and FUT4/LeY downregulation in vitro and in vivo [62]. Altogether, these studies suggest that suppression of FUT4 expression/activation by Rg3 may be a potential therapeutic strategy for melanoma treatment.

Agrawal and colleagues performed a systematic analysis of the melanoma glycome of clinical samples and found upregulation of core fucosylation (FUT8) and downregulation of α-1,2 fucosylation (FUT1, FUT2) as features of metastatic melanoma [19]. The pro-metastatic role of FUT8 was confirmed in vitro and in vivo. Indeed, FUT8 silencing decreased cell invasion and in vivo melanoma metastasis, suppressing the ability of melanoma cells to colonize distant organs. In addition, suppression of FUT8 impaired the growth of established metastasis in vivo using a model based on intracardiac instillation of tumor cells. The increase in FUT8 expression in melanoma cells was found to be transcriptionally regulated by TGFβ-induced factor homeobox 2 (TGIF2) [19]. Proteomic analysis of core-fucosylated proteins identified several regulators of invasion and metastasis, including the L1CAM. This cell adhesion molecule is cleaved by plasmin, and the proteolytic cleavage inhibits its ability to mediate spreading and metastasis [63]. FUT8-mediated core fucosylation prevents L1CAM cleavage by plasmin, facilitating melanoma cell invasion. The ability of uncleaved L1CAM to interact with the vasculature at distal organs may explain how FUT8 contributes to melanoma metastases [19] (Figure 1).

A recent report showed that fucosylation enhances the homing of antigen-specific cytotoxic T lymphocytes (CTL) to malignant niches, resulting in increased antitumor efficacy in hematological malignancies and solid tumors, including melanoma [64]. Authors showed that ex vivo expansion of T cells in fucosylation solution containing FUT-VII increases in vitro homing and cytotoxicity of antigen-specific CTLs. Furthermore, fucosylation enhanced in vivo CTL homing to melanoma in NOD/SCID and in immunocompetent C57BL/6 mice inoculated with B16F10 murine melanoma cells, boosting the antitumor activity of the antigen-specific CTLs. Importantly, this work demonstrates that fucosylation does not alter the specificity of the antigen-specific CTL for their targets and does not increase homing of CTLs to normal mouse tissue. This study provides a proof of principle for ex vivo CTL fucosylation as a novel approach to enhance the efficacy of adoptive cellular therapy in solid tumors, including melanoma [64].

The fucose salvage pathway is a two-step process in which mammalian cells transform L-fucose into GDP-L-fucose, a universal fucose donor used by fucosyltransferases to modify glycans [55]. It has been reported that the expression of fucokinase (FUK), a key enzyme in the fucose salvage pathway, is downregulated in metastatic melanoma, limiting GDP-L-fucose substrate availability [65]. Treatment of melanoma cells with L-fucose, or FUK overexpression, resulted in decreased migratory potential as well as increased cell surface fucosylation. Administration of L-fucose not only slowed tumor growth, but also inhibited lung metastases in a melanoma mouse model [65]. A recent report showed that the fucose salvage pathway inhibits invadopodia formation and extracellular matrix degradation by promoting α-1,2 fucosylation [66]. Activation of the fucose salvage pathway by ectopically expressed FUK decreased invadopodium numbers and inhibited the proteolytic activity of invadopodia in melanoma cells. Interestingly, the inhibition of invadopodium formation by L-fucose or FUK can be rescued by treatment with α-1,2, but not α-1,3/α-1,4 fucosidase, suggesting an α-1,2 fucose linkage-dependent anti-metastatic effect. Consistent with these results, the expression of the α-1,2 fucosyltransferase FUT1 was found downregulated during melanoma progression, and its ectopic expression was sufficient to inhibit invadopodium formation and extracellular matrix degradation in melanoma cells [66].

Altogether, these reports suggest that the functional consequences of fucosylation in melanoma are likely linkage-dependent. Indeed, the core-fucosylation (α-1,6 fucosylation) mediated by FUT8 promotes melanoma progression [19], whereas branched fucosylation through the α-1,2 linkage inhibits melanoma invasion and progression [19,65,66].

## 5. N- and I-Glycan Branching

Increased expression of complex β1,6-branched N-linked glycans represents a frequent glycosylation modification occurring in the Golgi membrane during malignant transformation and the acquisition of metastatic potential [67,68]. The enzymatic activity of α-1,6-mannosylglycoprotein 6-β-*N*-acetylglucosaminyltransferase V (GnT-V), which is encoded by the MGAT5 gene, produces tri/tetra-antennary *N*-glycan species that can modulate protein’s half-life, stability, extracellular-binding proteins as well as functional activity. Expression of MGAT5/GnT-V is transcriptionally regulated by several oncogenic inputs, including the transcription factor Ets-1 in several cancer cell lines including melanoma [69,70] and by the receptor tyrosine kinases Her-2/neu [71] and Src [72]. GnT-V expression can regulate N-glycosylation of Her-2 and Her-2-induced signaling pathways. Knockdown of GnT-V results in inhibited expression of N-linked β-1,6 branching on Her-2 and impaired Her-2-induced signaling pathways, which leads to upregulation of the protocadherin β gene cluster, contributing to reduced Her-2-mediated mammary tumorigenesis [73].

One of the first pieces of evidence that GnT-V is involved in melanoma progression came from the finding that hybrids produced by in vitro fusion of normal macrophages with Cloudman S91 melanoma cells display increased GnT-V activity, β1,6 branching in glycoproteins, upregulation of integrin subunits α3, α5, α6, αv, β1 and β3, and increased metastatic potential in vivo and motility in vitro [74,75]. Furthermore, *N*-glycoproteins bearing GlcNAc β1,6-branched N-glycans were identified in the metastatic A375 human melanoma cell line. These include integrin subunits α2, α3, α5 and β1, as well as N-cadherin and lysosome-associated membrane proteins (LAMP-1 and LAMP-2) [76]. A further study identified proteins bearing β1-6 branched N-glycans in human melanoma cell lines from different progression stages. Mass spectrometry analysis showed that primary melanoma cells WM35 show the lowest number of proteins possessing β1,6 GlcNAc branched N-glycans in comparison to metastatic WM9 and WM239 cell lines. Among identified proteins, the largest group consists of integrin subunits. In addition, L1CAM, Mac-2 binding protein, melanoma cell adhesion molecule, intercellular adhesion molecule, melanoma-associated antigen, melanoma-associated chondroitin sulfate proteoglycan 4 and lysosome-associated membrane protein (LAMP-1) were also found [77]. The increasing amount of GlcNAc β1,6 glycans on α5β1 and α3β1 integrins in metastatic cells plays a role in integrin-dependent migration on fibronectin and likely contributes to their acquisition of metastatic ability [78]. Another study reported that β1,6-branched N-glycans affect FAK signaling in metastatic melanoma cells, enhancing FAK autophosphorylation on Tyr397 and resulting in enhanced migration on vitronectin [79]. Cell surface N-glycans with GlcNAc β1,6 branches have also been shown to contribute to uveal melanoma progression by enhancing cell motility [80]. This could be in part explained by a more abundant presence of GlcNAc β1,6-branched N-glycans and higher expression of MGAT5 in uveal compared to cutaneous melanoma cell lines, contributing to their ability to migrate in fibronectin [81].

In contrast to GnT-V, β-1,4-mannosylglycoprotein 4-β-*N*-acetylglucosaminyltransferase III (GnT-III), which is encoded by MGAT3, catalyzes the addition of bisecting *N*-acetylglucosamine (GlcNAc) *N*-glycans in a β1,4-linkage of the *N*-glycan to produce a bisecting GlcNAc structure. The presence of the bisecting GlcNAc, which causes a conformational change in the glycan, results in the suppression of further processing and elongation, preventing the formation of highly branched species [82]. In this regard, GnT-III is mostly considered a suppressor of malignancy. It has been shown that overexpression of GnT-III (MGAT3) in highly metastatic mouse melanoma B16 cells led to a decreased synthesis of β1,6-branching through the competition for substrate between GnT-V and the overexpressed GnT-III. More importantly, GnT-III suppressed lung metastases in mice without affecting tumor growth [24]. Other reports showed that GnT-III represses tumor metastasis through regulation of key glycoproteins, such as EGFR, integrins and cadherins [25,26] (Figure 1). More recently, a study investigated the *N*-glycosylation profile of membrane and secreted proteins in WM266-4 metastatic melanoma cells, providing evidence that GnT-III upregulation does not lead to a total abrogation of the formation of highly branched glycans but modifies these glycans by the introduction of a bisecting N-acetylglucosamine (GlcNAc), modulating their capacity to interact with carbohydrate-binding proteins such as plant lectins [83]. This work suggests that the role of GnT-III in cancer is complex and needs to be further investigated.

Recent findings highlighted the critical new role for blood group I-antigens (I-branched glycans) as emerging effectors of cancer progression [84]. Synthesis of I-branched glycans, Galβ1,4GlcNAc moieties linked in β1,6 linkage to galactose residues on fetal i-antigen, is initiated by the master I-branching enzyme GCNT2 [85]. This enzyme regulates the conversion of linear poly-LacNAcs normally expressed on fetal and cord blood cells to I-branched glycans found on adult erythrocytes and mucosal epithelia [86,87]. GCNT2/I-branched glycans have been shown both positive and negative relationships with cancer progression, depending on the tumor type [84]. A recent study revealed that loss of GCNT2/I-branched glycans in melanomas regulates multiple cell surface glycoprotein signaling pathways and promotes melanoma growth and survival [22]. Sweeney and colleagues demonstrated that while normal epidermal melanocytes display abundant I-branches, these structures progressively diminish in primary and metastatic melanomas. This finding is in keeping with the inverse correlation between GCNT2 and melanoma progression reported by in silico and immunohistochemical analysis, suggesting that loss of GCNT2 expression could be used as a biomarker of melanoma [88]. Furthermore, this study clearly showed that knockdown of GCNT2 significantly enhances melanoma xenograft growth and three-dimensional colony formation and survival, whereas GCNT2 overexpression has the opposite effect. Analyses of two representative N-glycosylated protein families, insulin-like growth factor-1 receptor (IGF1R) and integrins, revealed that GCNT2/I-branched glycan modifications inhibited IGF-1 and ECM-mediated melanoma cell proliferation, survival and associated downstream signaling pathways [22] (Figure 1).

Thus, it is clear that N- and I-glycan branching are emerging as critical effectors of melanoma and correlate both positively and negatively with melanoma progression, regulating malignant-associated adhesive, migratory, growth, survival and metastatic activities. Table 1 summarizes the role of sialyltransferases, fucosyltransferases and N- and I-glycan branching enzymes in melanoma, and reports the identified substrates for each glycosyltransferase.

## 6. Conclusions and Future Perspectives

In this review, we have described the crucial role of aberrant glycosylation in melanoma progression. Some of these alterations may facilitate metastasis formation and immunosuppression during tumor progression, thus influencing the response to current therapies. A wide range of alterations in glycosylation has been reported in the late stages of melanoma progression. To exploit glycosylation, it will be critical to fully profile also early-stage melanomas and determine how glycosylation changes between the different stages.

It is clear from the presented literature that a more in-depth evaluation of glycosylation changes occurring in melanoma is required for further mechanistic understanding of the disease. However, given the structural complexity of glycan structures and the heterogeneity in glycosylation sites, a complete characterization of tumor glycomics and glycoproteomics represents a challenge. Cancer glycomics is currently performed on total cell preparations using liquid chromatography and mass spectrometry analysis, which enables the detailed analysis of all the structural glycans of cancer cells. However, the use of lectins and antibodies is the most common way to analyze glycosylation of cells, given the complexity of chemical methods. Therefore, the development of new and more accessible techniques for the determination of glycosylation may help to extent the use of this type of analysis. Furthermore, the creation of a glycobiology-focused melanoma database that examines not only the expression of specific enzymes but also the presence of gain-of-function or loss-of-function mutations and splicing variants of glycoproteins and enzymes could be very useful for glycomic research in melanoma.

The rationale of targeting aberrant glycosylation in melanoma is supported by several studies discussed in this review. For instance, two recent studies identified core fucosylation (FUT8) and the sialyltransferase ST3GAL1 as critical drivers of melanoma metastasis, highlighting the therapeutic potential of targeting FUT8 and ST3GAL1 to treat metastatic melanoma [19,20]. Therefore, these studies provide a rationale for the future design of small molecule inhibitors against FUT8 and ST3GAL1 to prevent or treat established melanoma metastasis. However, this field of research is still very challenging. To proceed from pre-clinical to clinical trial, it is critical that the inhibitors are selective for the specific glycosyltransferase to reduce off-target effects on the liver and kidney. In addition, the development of clinically relevant glycosyltransferase inhibitors as potential anti-cancer treatments requires good cell permeability and bioavailability. To date, the kinetics of these enzymes coupled with information derived from structural studies allowed the design of selective catalytic glycosyltransferase inhibitors that have been successfully tested in pre-clinical studies. Other strategies include targeting altered cancer-associated glycans, for instance with carbohydrate analogs [89] or with glycan-specific CAR-T [90].

Another intriguing option is represented by the therapeutic manipulation of glycosylated targets. For instance, the glycosylation inhibitor per-*O*-acetylated GlcNAcβ1,3Galβ-*O*-naphthalenemethanol (AcGnG-NM), which inhibits the biosynthesis of sLe^X^ in tumor cells [91,92,93,94], has been evaluated in vivo. Systemic administration of AcGnG-NM significantly inhibited dissemination of murine Lewis lung carcinoma to the lungs without affecting the level of circulating leukocyte or platelets [95]. Because AcGnG-NM inhibits the formation of selectin ligands on several types of cancer cell lines in vitro [91,92,93,94], it will be worth evaluating this and other disaccharides as adjuvant therapy for blocking metastatic spreading in melanoma.

Recently, Shi and colleagues have emphasized the translational value of targeting protein glycosylation itself, offering new clinical perspectives for tumor immunotherapy [96]. In this study, the authors identified MAN2A1, which encodes a glycosyl hydrolase involved in the complex N-glycan biosynthesis, as a functional immunoregulator. Genetic and pharmacologic inhibition of MAN2A1 with swainsonine sensitized tumors to anti-PD-L1 treatment, providing a survival benefit in a syngeneic B16F10 mouse model. Consistently, low expression of MAN2A1 is associated with a better prognosis with improved cytotoxic tumor infiltrating lymphocyte signatures upon anti-PD-L1 treatment [96]. Thus, targeting protein glycosylation alone or in combination with immunotherapy or targeted therapy could improve the prognosis of metastatic melanoma [97].

In conclusion, in this review, we summarized the changes in glycosylation involved in melanoma progression and provided inputs to exploit aberrant glycosylation, sialylation, fucosylation and glycan branching with the aim of finding alternative therapeutic options for melanoma treatment. We believe that the knowledge of the melanoma “glycome” and its critical regulatory effects on melanoma biology and progression, as well as the interplay of the tumor with the immune system, will set the basis for novel and more effective therapeutic strategies against this dismal disease.

## Figures and Tables

**Figure 1 cells-10-02136-f001:**
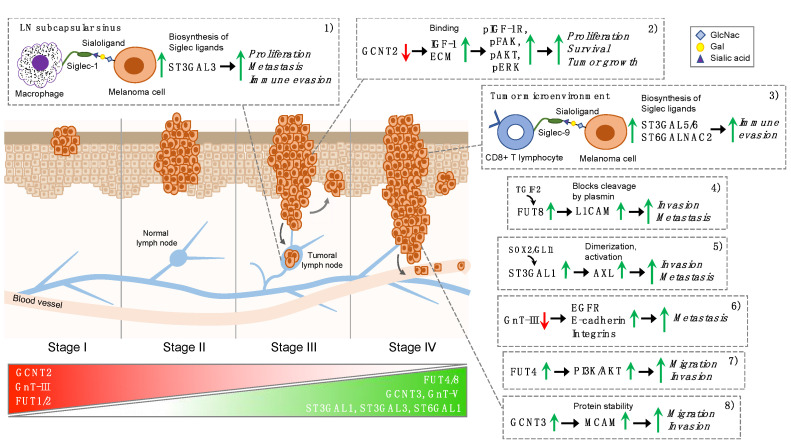
Glycosyltransferases as emerging effectors of melanoma progression. Schematic representation of selected glycosyltransferases and their role in the different stages of melanoma progression, based on the American Joint Committee on Cancer (AJCC) staging system. Stage I melanoma is no more than 1 mm thick; stage II extends beyond the epidermis into the thick dermis layer of the skin; stage III has metastasized to nearby lymph nodes, lymph vessels or skin; stage IV has metastasized to other places throughout the body, such as the brain, lungs or liver. Expression levels of FUT1/2, GCNT2 and GnT-III are high in early-stage and diminish during melanoma progression, whereas expression of FUT4/8, GCNT3, GnT-V, ST3GAL1, ST3GAL3 and ST6GAL1 are higher in late-stage melanomas. Glycosyltransferases may promote melanoma progression by several mechanisms: (**1**) interaction between Siglec1-expressing macrophages and metastatic cells facilitates metastatic colonization [21]; (**2**) loss of GCNT2 enhances growth factor receptor and integrin-mediated cell proliferation signaling pathways promoting melanoma growth and survival [22]; (**3**) Siglec-9/sialoligand interactions result in a tumor glycosylation-dependent circuit that suppresses CD8+ T cell effector responses in the tumor microenvironment [23]; (**4**) FUT8-mediated core fucosylation prevents L1CAM cleavage by plasmin, facilitating melanoma cell invasion [19]; (**5**) ST3GAL1 promotes melanoma metastasis through AXL [20]; (**6**) GnT-III represses metastasis through regulation of EGFR, integrins and cadherins [24,25,26]; (**7**) FUT4 regulates melanoma cell migration and invasion through activation of the PI3K/AKT signaling pathway [27]; (**8**) GCNT3 promotes melanoma cell migration and invasion through stabilization of MCAM [28]. See text for details regarding the role of the reported glycosyltransferases. Abbreviations used in the figure are listed in the “Abbreviation” section.

**Table 1 cells-10-02136-t001:** The role of glycosyltransferases in melanoma.

Glycosyltransferase	Gene	Target	Effects	References
β-galactoside α2,3-sialyltransferase 1	ST3GAL1	AXL	Cell invasion Metastasis	[20]
β-galactoside α2,3-sialyltransferase 3	ST3GAL3	-	Cell proliferation Metastasis Immune evasion	[21]
β-galactoside α2,3-sialyltransferase 5	ST3GAL5	-	Immune evasion	[23]
β-galactoside α2,3-sialyltransferase 6	ST3GAL6	-	Immune evasion	[23]
β-galactoside α2,6-sialyltransferase 1	ST6GAL1	-	Cell invasion	[45]
*N*-acetylgalactosaminide α2,6-sialyltransferase 2	ST6GALNAC2	-	Immune evasion	[23]
α-1,3-fucosyltransferase 4	FUT4	PI3K/AKT	Cell migration and invasion	[27]
α-1,6-fucosyltransferase	FUT8	L1CAM	Cell invasion Metastasis	[19]
α-1,6-mannosylglycoprotein 6-β-N-acetylglucosaminyltransferase (GnT-V)	MGAT5	α5β1, α3β1 integrins	Cell migration	[78,79]
β-1,3-galactosyl-*O*-glycosyl-glycoprotein β-1,6-*N*-acetylglucosaminyltransferase 3	GCNT3	MCAM	Cell migrationand invasion	[28]
β-1,4-mannosylglycoprotein 4-β-*N*-acetylglucosaminyltransferase (GnT-III)	MGAT3	EGFRE-cadherins Integrins	Metastasis suppression	[24,25,26]
*N*-acetyllactosaminide β-1,6-*N*-acetylglucosaminyltransferase	GCNT2	IGF-1ECM proteins	Tumor growth inhibition	[22]

## Data Availability

Not applicable.

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
