# Peer review of "The Role of Glycosylation in Melanoma Progression"

_cells, 2021, doi:10.3390/cells10082136_

Round 1

Reviewer 1 Report

This paper is well written and worth publishing.  Before publication,however, references cited should be corrected and some references should be added.

1) Ref. 64  the authors did not mention anything about signaling pathway. Implication of ets and her-2 etc, were reported by other groups even though cancer cell lines are different.   (Kang, JBC 1996, Ko et al. JBC 274,1999, Chen et al. ,Oncogene, 1998, Bruckhaults et al.  JBC 272,1997 , Guo et al, JBC 287,2012)

2) Chakraborty AK and their groups reported earlier about the role of GnT-V in melanoma and macrophage hybrid.  These papers should be cited. 

Author Response

This paper is well written and worth publishing.  Before publication, however, references cited should be corrected and some references should be added.

Response: We thank the Reviewer for the very positive comments on our manuscript. We fixed references as requested (see below).

Point 1) Ref. 64  the authors did not mention anything about signaling pathway. Implication of ets and her-2 etc, were reported by other groups even though cancer cell lines are different. (Kang, JBC 1996; Ko et al. JBC 274,1999; Chen et al., Oncogene; 1998, Bruckhaults et al. JBC 272,1997; Guo et al, JBC 287, 2012).

Response: We apologize for the mistake regarding Ref. 64 (now Ref. 63), we now corrected it in the text (line 396). We also thank the Reviewer for suggesting a number of references regarding the transcriptional regulation of GnT-V by several inputs and signaling pathways. All the suggested studies have been added in the text and reference list (Ref. 65 to 69, lines 400-407 in the text).

Point 2) Chakraborty AK and their groups reported earlier about the role of GnT-V in melanoma and macrophage hybrid.  These papers should be cited. 

Response: We thank the Reviewer for the suggestion. In the revised version of the manuscript we added two studies from Chakraborty  and his group about the role of GnT-V in melanoma and macrophage hybrids (Ref. 70 and 71, lines 408-412 in the text).

Reviewer 2 Report

The authors provide a clear and updated overview of the changes in glycosylation involved in melanoma progression, and provided inputs to exploit aberrant glycosylation, sialylation, fucosylation and glycan branching with the aim to find alternative therapeutic options for melanoma treatment. The review can help readers get the overall knowledge of the alteration and role of glycosylation in the melanoma and its critical regulatory effects on the interplay of the tumor with the immune system, suggesting more further researches to investigate pathogenesis of melanoma and find novel and more effective therapeutic strategies against melanoma. It is worthy to publish after minor revision as follows: 1) The resolution of figure 1 need improved. 2) Add the major reference along with each change arrow in figure 1 will be useful. 3) That will be more useful if the authors can address more clearly about their opinions about what is the major challenge in the more in-depth evaluation of glycosylation changes occurring in melanoma as well as finding effective therapeutic strategies and provide some of their hypotheses based on previous research work.

Author Response

The authors provide a clear and updated overview of the changes in glycosylation involved in melanoma progression, and provided inputs to exploit aberrant glycosylation, sialylation, fucosylation and glycan branching with the aim to find alternative therapeutic options for melanoma treatment. The review can help readers get the overall knowledge of the alteration and role of glycosylation in the melanoma and its critical regulatory effects on the interplay of the tumor with the immune system, suggesting more further researches to investigate pathogenesis of melanoma and find novel and more effective therapeutic strategies against melanoma. It is worthy to publish after minor revision as follows.

Response: We thank the Reviewer for the very positive comments on our manuscript.

Point 1) The resolution of figure 1 need improved.

Response: The resolution of Figure 1 has been improved.

Point 2) Add the major reference along with each change arrow in figure 1 will be useful.

Response: We thank the Reviewer for the suggestion. We numbered each single diagram in Figure 1 (1-8) and added the major references in Figure legend. We also briefly described the role of the selected glycosyltransferases in promoting melanoma progression in figure legend (lines 113-122).

Point 3) That will be more useful if the authors can address more clearly about their opinions about what is the major challenge in the more in-depth evaluation of glycosylation changes occurring in melanoma as well as finding effective therapeutic strategies and provide some of their hypotheses based on previous research work.

Response: We thank the Reviewer for the suggestion. In the revised version of the manuscript we added additional discussion about the major challenge in the evaluation of glycosylation changes in melanoma (lines 494-505) and finding effective therapeutic strategies based on previous work (lines 506-516).